# Ribose and Non-Ribose A2A Adenosine Receptor Agonists: Do They Share the Same Receptor Recognition Mechanism?

**DOI:** 10.3390/biomedicines10020515

**Published:** 2022-02-21

**Authors:** Giovanni Bolcato, Matteo Pavan, Davide Bassani, Mattia Sturlese, Stefano Moro

**Affiliations:** Molecular Modeling Section (MMS), Department of Pharmaceutical and Pharmacological Sciences, University of Padova, 35131 Padova, Italy; giovanni.bolcato.1@phd.unipd.it (G.B.); matteo.pavan.7@phd.unipd.it (M.P.); davide.bassani.1@studenti.unipd.it (D.B.); mattia.sturlese@unipd.it (M.S.)

**Keywords:** adenosine receptors, non-nucleoside agonists, supervised molecular dynamics (SuMD), water molecules

## Abstract

Adenosine receptors have been a promising class of targets for the development of new therapies for several diseases. In recent years, a renewed interest in this field has risen, thanks to the implementation of a novel class of agonists that lack the ribose moiety, once considered essential for the agonistic profile. Recently, an X-ray crystal structure of the A_2A_ adenosine receptor has been solved, providing insights about the receptor activation from this novel class of agonists. Starting from this structural information, we have performed supervised molecular dynamics (SuMD) simulations to investigate the binding pathway of a non-nucleoside adenosine receptor agonist as well as one of three classic agonists. Furthermore, we analyzed the possible role of water molecules in receptor activation.

## 1. Introduction

Adenosine is the endogenous agonist of a group of class A G protein-coupled receptors (GPCRs) named adenosine receptors (AR); four receptors belong to this family: A_1_, A_2A_, A_2B_, and A_3_. A_1_ and A_3_ are generally coupled to the G_αi_ protein (therefore, they inhibit the adenylate cyclase enzyme upon activation); A_2A_ and A_2B_ are coupled to the G_αs_ protein (and, therefore, stimulate the production of cAMP upon activation) [1].

AR are targets of interest for the treatment of several diseases [2]: Parkinson’s disease [3,4], asthma [5], pain treatment [6], several cancer types [7], and cardiovascular diseases [8]. Despite this broad range of potential therapeutic applications, only two AR ligands have been approved: theA_2A_ antagonist Istradefylline, approved for the treatment of Parkinson’s disease, and the A_2A_ agonist Regadenoson, used as a coronary vasodilator. One of the main problems in the translation of AR ligands into therapeutic agents is the presence of unacceptable side effects due to the lack of selectivity of the drug candidates among different AR subtypes as well as off-target effects [9].

Progress was made in the field of AR agonists as therapeutic agents [10] with the publication by Bayer of some patents regarding non-nucleoside AR agonists [11]. While this novel class of AR agonists presents several advantages over classic adenosine-derived ligands (easier synthesis, improved pharmacokinetics, and oral bioavailability), the AR activation for this class of compounds has been difficult to understand since they lack the ribose moiety, which was considered essential for the agonistic profile of AR ligands [12]. Some modifications on this moiety are tolerated and can improve both metabolic stability and potency, but often, these alterations on the ribose unit lead to inactive compounds or switch the ligand activity toward an antagonistic profile [13].

To gain some insights on the structural basis of AR activation, several site-directed mutagenesis data have been collected over the years [14]. Interestingly, it was proven that the non-nucleoside A_2A_ agonist LUF5834 is sensitive to mutagenesis experiments in a different way compared to classic adenosine-like AR agonists such as CGS21680 [15]. In particular, the agonistic profile of LUF5834 is not affected when Ser-277 and Thr-88 are mutated in alanine (note that the enumeration, as well as in all this work, refers to A_2A_). These two residues are essential for the agonistic activity of classic AR agonists along with His-278.

Fundamental progress in the comprehension of the structural basis of the agonistic action of non-nucleoside agonists has been made recently with the obtainment of the X-ray crystal structure of the A_2A_ AR in the complex with a close analog of LUF5834, LUF5833 [16]. Interestingly, the ligand does not form any hydrogen bonds with the abovementioned residues that are considered essential for the activation of A_2A_ AR (Thr-88, Ser-277, His-278). In Figure 1, a comparison of the binding mode (as observed in X-ray crystal structures) that the endogenous agonist adenosine and of LUF5833 adopt in the orthosteric binding site of A_2A_ AR is reported.

Starting from this structural information, in the present work we have investigated the recognition process of LUF5833 and different classic adenosine-like agonists: CGS21680, NECA, and adenosine itself. The study has been carried out using supervised molecular dynamics (SuMD) simulations to gain structural information beyond the observed final bound state. SuMD is, indeed, a molecular dynamics-based approach that allows the sampling of events involving infrequent particle collisions, such as protein-ligand binding, without applying any energetic bias to the system.

The comparison between the binding trajectories collected for the two different classes of AR agonists reveals a different recognition pathway. Moreover, a detailed analysis of the behavior of water molecules during the binding event provides some insights into the possible role of the solvent molecules in the activation of the A_2A_ adenosine receptor.

## 2. Materials and Methods

### 2.1. System Setup

The three-dimensional structure of the protein−ligand complexes examined in this work (PDB codes: 2YDO, 2YDV, 4UG2, and 7ARO) was retrieved from the Protein Data Bank (PDB) and prepared for subsequent calculations using various tools provided by the Molecular Operating Environment (MOE) suite, version 2019.01 [17]. Residues with alternate conformation were assigned to the one with the highest occupancy. Missing hydrogen atoms were added to the system with the Protonate3D tool, assigning each titratable residue to the most probable protonation state at pH = 7.4. Crystallographic water molecules, ions, and other molecules present in the crystallization buffer were then removed, and the ligand was moved away from the binding site into the bulk, at a distance of at least 30 Å from the nearest receptor atom (higher than the cutoff chosen for electrostatic interaction computation).

The system preparation for the supervised molecular dynamics (SuMD) simulations was carried out with Visual Molecular Dynamics (VMD), version 1.9.3 [18]). At first, the protein−ligand system was explicitly solvated in a cubic TIP3P [19] water box, ensuring a distance of 15 Å between the box borders and any protein atom. Then, the system charge was neutralized by the addition of sodium and chlorine ions until a physiological concentration (0.154 M) was reached. Finally, the receptor was embedded in a lipid bilayer consisting of phosphatidylcholine (POPC) units.

From a methodological point of view, one main limitation of the SuMD technique, as is the case for traditional molecular dynamics (MD), is the fact that simulations are carried out assuming fixed protonation states. The prediction of the protonation state of titratable residues relies on a static structure (the crystal complex, which is the starting point for the simulations) and can sometimes be imprecise in those cases where the protein is flexible [20] or the residues are highly buried [21]. Furthermore, the coexistence of protonated and deprotonated states and dynamical processes coupled to a change in protonation states cannot be directly studied if the protonation states are fixed.

A second limitation is represented by the fact that the lipid constitution of the phospholipid membrane does not include the presence of cholesterol, which could exert some form of allosteric modulation on the AR [22].

### 2.2. Molecular Dynamics

All simulations were carried out using the ACEMD [23] molecular dynamics engine. The system was described using parameters from the CHARMM36 [24] force field (protein, lipids, ions, and water molecules) while ligand parameters were retrieved from Paramchem [25], a web front-end for the CGenFF [26] force field. If the parameters associated with specific dihedral angles of ligands presented high penalties, these have been parametrized using FFParam [27]. A QM scansion of the dihedral angle has been performed using the MP2 level of theory with the 6-31G** basis set; then, the QM profile has been fitted to retrieve the new force field parameters.

The simulation protocol consisted of a four-stage equilibration phase, followed by a productive SuMD simulation phase. For both equilibration and productive simulations, the integration timestep was set to 2 fs and the temperature was set to 310 K through a Langevin thermostat (friction coefficient = 0.1 ps^−1^); the M-SHAKE algorithm was employed to constrain the length of bonds involving hydrogen atoms, and the particle mesh Ewald (PME) [28] was exploited to compute electrostatic interactions (grid length = 1 Å). Finally, a 9.0 Å cutoff was applied to long-term interactions.

### 2.3. Equilibration Phase

Before equilibration MD simulations, 1500 steps of energy minimization, using the conjugate gradient method, was performed to remove clashes and bad contacts within the system. The first three equilibration MD simulations were carried out in the isothermal−isobaric ensemble (NPT), maintaining the system pressure fixed at 1 atm through the Berendsen barostat [29] while the fourth and final one was performed in the isothermal ensemble (NVT).

The first equilibration stage consisted of a 5 ns simulation with 1 kcal mol^−1^ Å^−2^ harmonic positional constraints applied on each receptor, ligand, and membrane atom. The second equilibration stage consisted of a 10 ns simulation with the same constraints applied only on each protein, ligand, and phosphorus atom. The third equilibration stage consisted of a 5 ns simulation with the same constraints applied only on the protein alpha carbons and on ligand atoms. Finally, a 10 ns equilibration MD simulation was performed without any constraints applied to the system.

### 2.4. Supervised Molecular Dynamics (SuMD) Simulations

SuMD [30] is an enhanced sampling MD method that allows investigating molecular recognition processes at an atomistic level of detail in the nanosecond timescale without any energetic bias. The SuMD code is written in Python 2.7 and mainly exploits the Numpy and ProDy [31] packages to perform geometric supervision over a series of short classic MD trajectories (defined as “SuMD steps”) carried out with the ACEMD engine. As reported in the original publication, each SuMD step lasts 600 ps.

During each SuMD step, the distance between the center of mass (i.e., the hypothetical point where the entire mass of an object is assumed to be concentrated) of both the ligand and the binding site is monitored and collected at 5 evenly spaced time intervals. At the end of each step, these data are fitted in a straight line, which is then processed by a tabù-like algorithm: if the line slope is negative (indicating that the ligand is approaching the binding site), the step is considered productive and retained for the generation of the final MD trajectory while the final state of this simulation is set as the initial state for the successive step. On the contrary, in the case where the slope is positive (indicating that the ligand is not approaching the binding site), the step is considered not productive and is discarded; in this case, the step is repeated, reassigning the velocities through the Langevin thermostat. This process continues until the distance between the two centers of mass drops below 5 Å: from that point on, the supervision is turned off, and the simulation proceeds as a classic MD simulation for the other 30 SuMD steps.

### 2.5. Trajectory Analysis

A per-residue energetic analysis was performed using an in house-developed Python script.

At first, the MD Analysis [32,33] Python package is exploited to parse each MD trajectory and compute the number of contacts between the ligand and each protein residue, using a cutoff distance of 4.5 Å.

Afterward, the interaction energy (defined as the sum of the electrostatic and van der Waals contribution) is computed between the ligand and each one of the top 25 most contacted residues alongside each MD trajectory using the NAMD Energy Plugin [34] for VMD (version 1.4).

Finally, a heatmap is generated, exploiting the Seaborn Python package: on the horizontal axis, the simulation time in nanoseconds is reported while on the vertical axis, the residue name and index are reported for each residue considered for this analysis. The interaction energy is then plotted onto the heatmap using a colormap which ranges from red (indicating positive energy values, i.e., a repulsive interaction) to blue (indicating negative energy values, i.e., an attractive interaction). The first and the third quartile, with regards to the distribution of interaction energy values, are used as mask values for the heatmap generation.

To inspect the peculiar hydrodynamic profile of ligand LUF5833, the trajectory was analyzed with AquaMMapS [35], an in house-developed tool that allows investigating the behavior of water molecules within a receptor, based on their persistency across an MD trajectory. For this purpose, the simulation box is discretized in a voxel grid, and the occupancy value for each cell is calculated as the ratio between the number of frames in which a water molecule occupies that cell and the total number of frames.

## 3. Results and Discussion

For all of the four ligands, a SuMD trajectory, where the observed crystallographic binding mode is well reproduced, has been collected (see also videos in the Appendix A). In Figure 2, the crystallographic binding mode and the final pose obtained with SuMD for the four ligands under examination are reported: as it can be seen, the experimentally observed binding mode is well reproduced.

A detailed analysis on each trajectory has been performed to understand the recognition process for the four agonists (Figure 3, Figure 4, Figure 5 and Figure 6). This analysis consists of a per-residue decomposition of the interaction energy between the ligand and the protein, during the binding event.

The binding pathway for the four ligands as well as the most contacted regions of the proteincan be visualized in Figure 7 and the videos collected in the Appendix A.

The trajectory of LUF5833 has been prolonged for 25ns at the end of the SuMD simulation. This prolonged trajectory has been analyzed using the AquaMMapS (see Section 2) to gain information on the possible role of the solvent in the activation mechanism of AR by non-nucleoside agonists.

For comparison, the first part (before the ligand reaches the orthosteric site) of the SuMD trajectory of LUF5833 has been analyzed using the AquaMMapS. This analysis can provide some additional information on the solvent behavior in the apo form of the receptor. The results of these analyses are reported in Figure 8.

As it can be observed, the three classic ribose-containing agonists approach the protein taking contacts in a region that includes residues of the extracellular loop (ECL) 2 and 3 and transmembrane helical segments (TMs) 4, 5, and 6. This meta-binding site has already been described in our previous works [36,37]. LUF5833, instead, approaches the receptor from the other side, making contacts in the region between the extracellular portion of TMs 1, 2, and 7. This suggests a different binding pathway for the two classes of AR agonists. Trajectory analysis correctly highlights the pivotal role that is played by Phe-168 (which is involved in a π-stacking with both types of agonists), according to mutagenesis studies that flagged this residue as fundamental for ligand binding [15]. Moreover, Asn-253,which establishes a double hydrogen bond with the adenine moiety of ribose agonists and a single hydrogen bond through one of the two nitrile groups of LUF5833, is also marked as an important residue for the recognition of both classes of agonists, according to mutagenesis data that illustrate how an N253A mutation would be detrimental for the activity of both ribose and non-ribose agonists [15].Aside from these common interaction features that regard the adenine or “sudo-adenine” portion of the molecule, the main difference in the recognition pattern of these two classes of agonists is related to the role of Ser-277 and Thr-88: as highlighted by our trajectory analysis, neither of these two residues establishes a direct interaction with LUF5833,in agreement with mutagenesis data, which show that mutation of these two residues negatively impact ribose agonists but have no effect on the affinity of the non-ribose one [15].

Regarding the solvent behavior in the orthosteric site, it is interesting to note that the water molecules in the apo form of the receptor seem to adopt an interactive pattern that mimics the one observed for agonists ligands. Indeed, key residues for the activation of the receptor, such as Thr-88, His-250, Ser-277, and His-278 are well solvated and stabilize water molecules through hydrogen bonds.

It is tempting to argue that this observation (the organization of solvent molecules in a way that mimics agonists interactions) can provide a possible explanation for the concept of receptor basal activity, defined as the activation of the receptor in the absence of the ligand.

In detail, it seems that the stable water molecule interacting with Ser-277 is displaced upon LUF5833 binding while the water molecule interacting with His-278 is further stabilized by the cyano group in position 3. This water molecule is displayed in Figure 8. Therefore, while it is true that LUF5833 does not interact directly with any key residues for the receptor activation, at least one of these interactions (the one with His-278) is still present and is mediated by a stable water molecule. Interestingly, the interaction between adenosine and His-250 is mediated by a bridging water molecule while NECA and CGS21680 interact with this residue using their amide tail.

Concerning water-bridged interactions, the AquaMMapS analysis illustrates how LUF5833 seems to stabilize two water molecules that form a hydrogen bond bridge between His-250 and Thr-88 (Figure 8), playing a similar role to the amide tail of both NECA and CGS. The mutation of both residues has a detrimental role on ribose agonists’ affinity, coherently with their direct interaction with the ribose moiety [38,39]. This could indicate that while not interacting with His-250 and Thr-88, non-ribose agonists, such as LUF5833, could stabilize a water molecule network that mimics the same interaction pattern of ribose agonists. The hydrophobic pocket which houses these stable water molecules is completed by Leu-85: this residue was determined to have a big impact on the affinity of ribose agonists such as CGS but has a smaller effect on the affinity of non-ribose agonists such as LUF5833 [15]. This could be explained by the fact that this residue interacts directly with the ligand in the case of CGS while in the case of LUF5833, its main involvement seems to be in the definition of a “hydrophobic” trap for these two water molecules that mimic the interaction pattern of ribose agonists. Notably, this bound water network extends also towards Asn-253: based on mutagenesis studies which show that, in the case of non-ribose agonists, the reduction of potency is mainly related to efficiency rather than on binding affinity, it is also tempting to speculate that this water network stabilized by LUF5833 is somehow involved in receptor activation, thereby validating the role of LUF5833 as a partial agonist.

Altogether, our SuMD simulations provide an overview of the mechanistic details regarding the recognition process between the AR and their agonists, shedding light upon differences in the binding event between nucleoside and non-nucleoside ones. Despite the useful information that can be gathered from our simulations, some AR-specific features cannot be captured by the SuMD technique, thereby impairing a clear and complete depiction of the agonist mechanism. Firstly, our simulations consider the interaction between one single ligand molecule and an individual receptor in a defined lipidic and ionic environment: despite being a sufficiently accurate approximation of reality for the evaluation of geometric properties related to the binding event, these boundary conditions cannot take into account the complex network of interactions of the AR within a cellular environment, including the ones with themselves, other GPCRs, and a plethora of ancillary factors [40], which leads to surprising pharmacological properties [41,42]. Secondly, a key aspect of the AR agonist signaling is portrayed by the ligand residence time, which has been flagged as a more efficient predictor of “in vivo” functional efficacy than binding affinity [43]. Although the evaluation of this aspect of agonist signaling was beyond the scope of this scientific work, it is important to underline that the association process is only the first part of a more complex and intricate story.

## Figures and Tables

**Figure 1 biomedicines-10-00515-f001:**
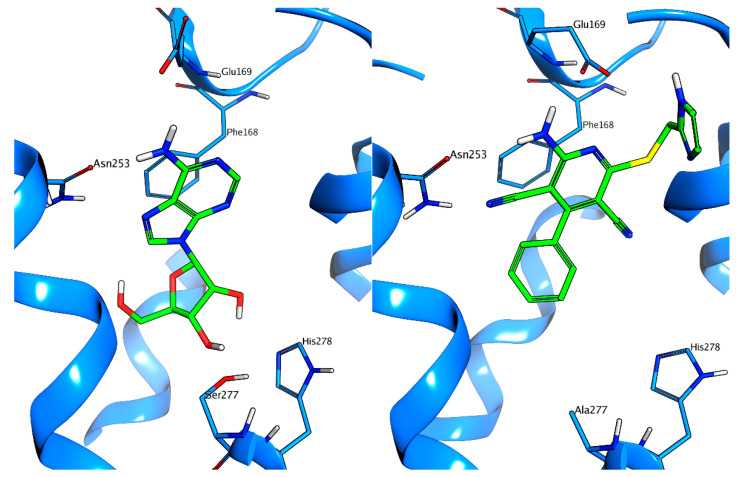
Binding modes of adenosine (**left**) and of LUF5833 (**right**) as observed in X-ray crystal structures (PDB code 2YDO and 7ARO, respectively). Please note that the AR used to obtain the crystal structure with LUF5833 present some thermostabilizing mutations, including Ser-277.Other AR agonists, such as CGS21680 and NECA, also interact with His-250 and Thr-88. The binding mode of these two ligands can be found in Figure 2.

**Figure 2 biomedicines-10-00515-f002:**
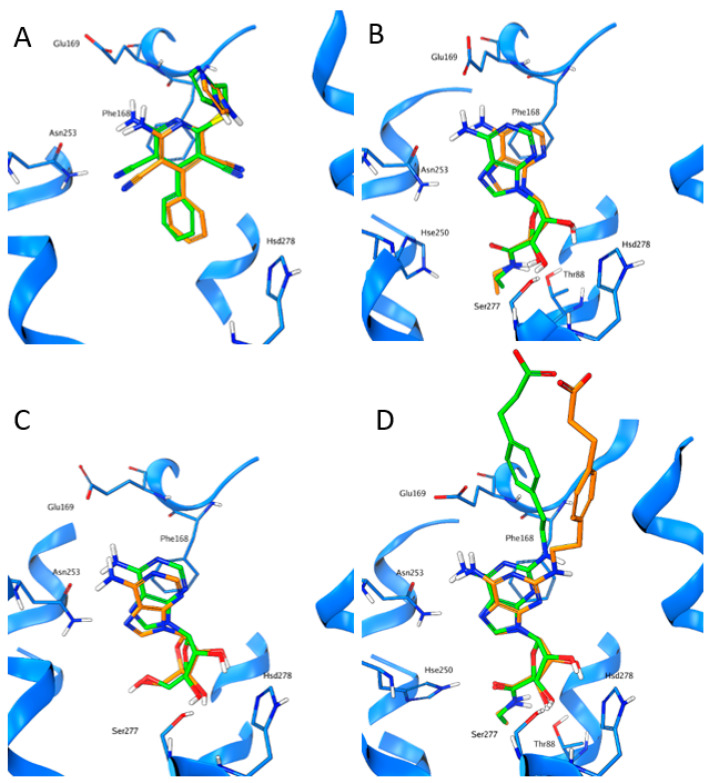
The crystallographic binding mode (green) and the final pose obtained using SuMD (orange) for the four agonists used in this study ((**A**): LUF5833; (**B**): NECA, (**C**): Adenosine; (**D**): CGS21680).

**Figure 3 biomedicines-10-00515-f003:**
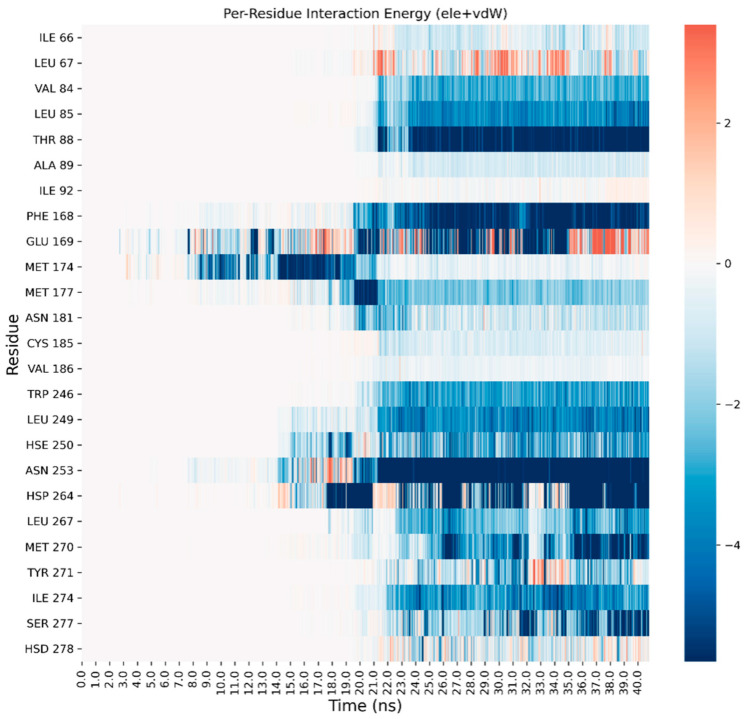
Per-residue energetic analysis of the SuMD trajectory of CGS21680.

**Figure 4 biomedicines-10-00515-f004:**
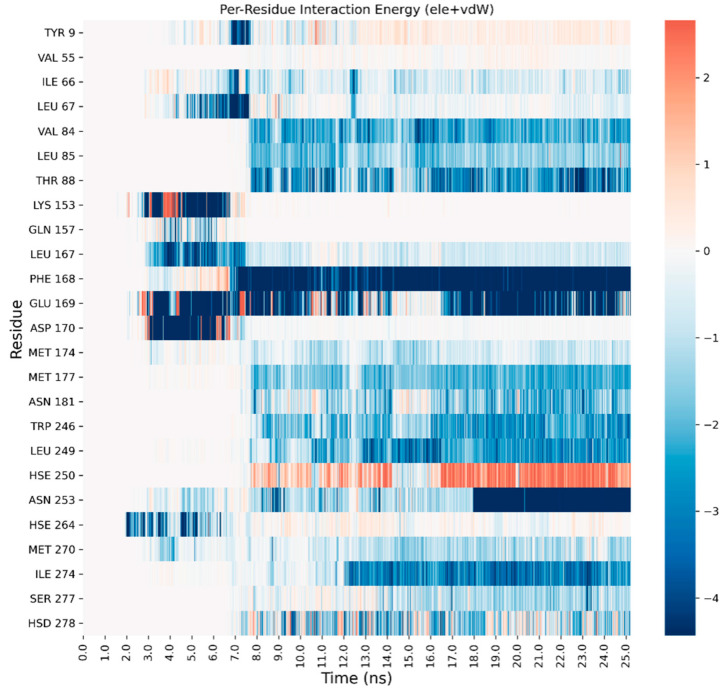
Per-residue energetic analysis of the SuMD trajectory of adenosine.

**Figure 5 biomedicines-10-00515-f005:**
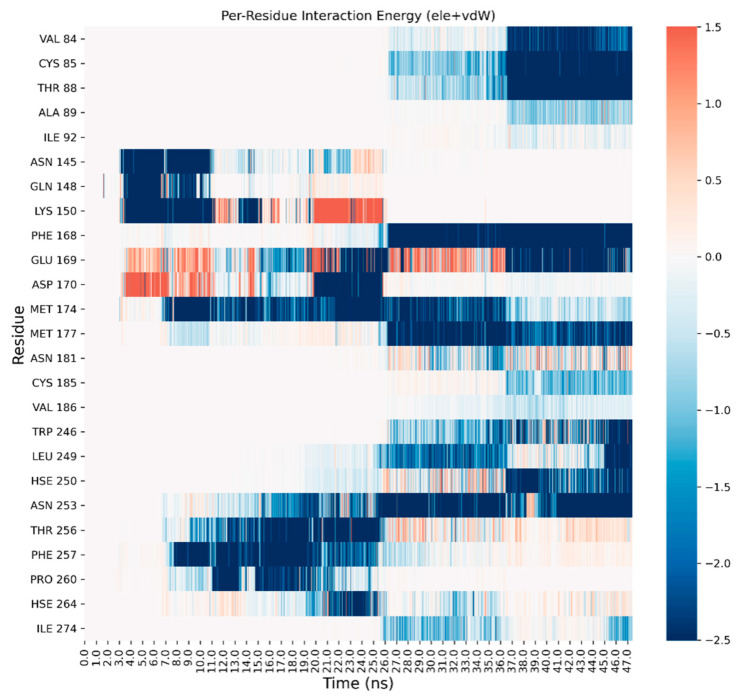
Per-residue energetic analysis of the SuMD trajectory of NECA.

**Figure 6 biomedicines-10-00515-f006:**
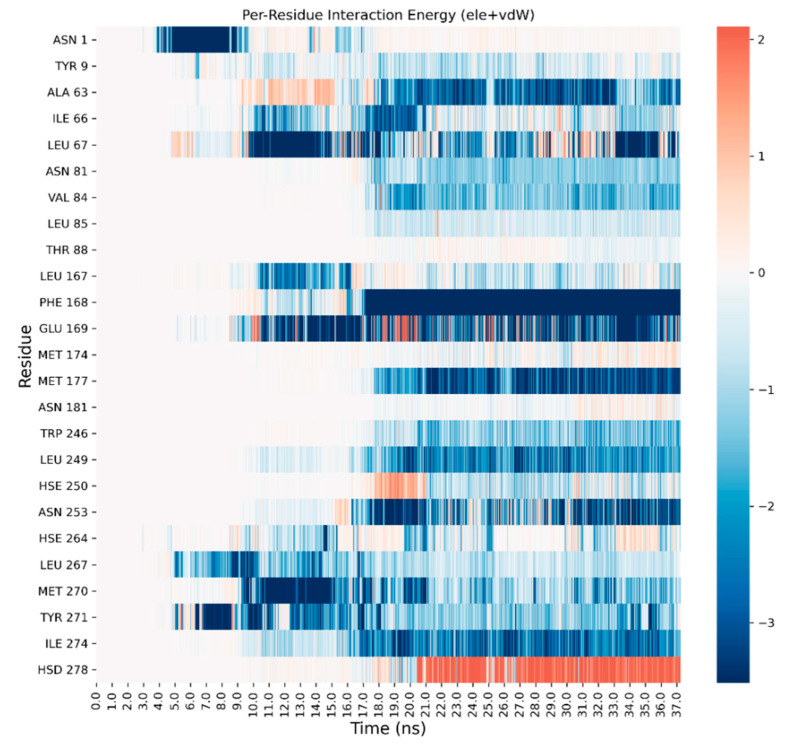
Per-residue energetic analysis of the SuMD trajectory of LUF5833.

**Figure 7 biomedicines-10-00515-f007:**
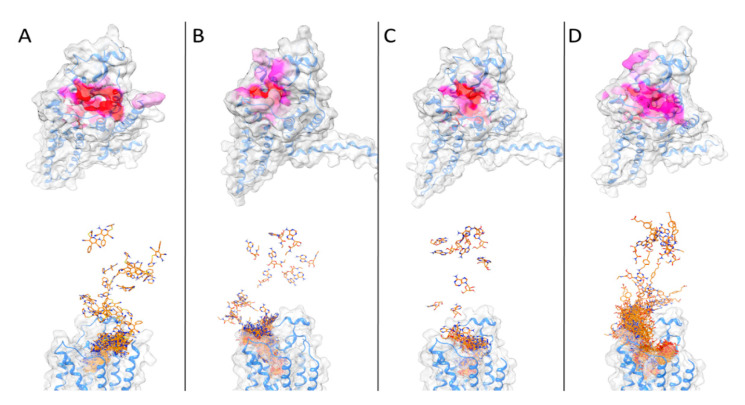
In the upper part of the figure, the protein surface is colored according to the number of contacts with the ligand during the trajectory (scale white to red, from less contacted to more contacted residues). In the lower part, the SuMD trajectory is displayed superposing each frame. (**A**): LUF5833; (**B**): NECA; (**C**): Adenosine; (**D**): CGS21680.

**Figure 8 biomedicines-10-00515-f008:**
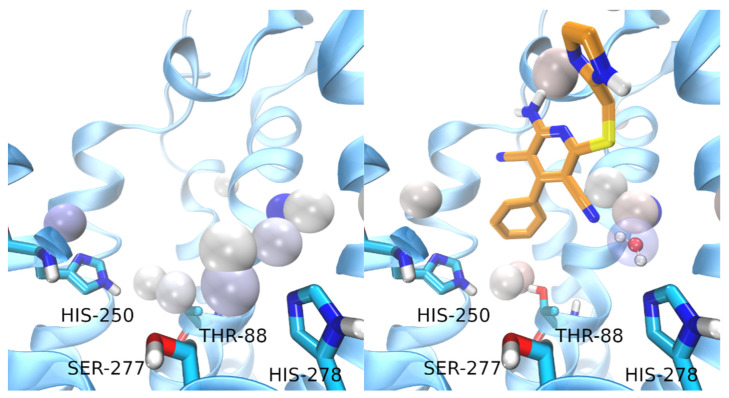
Results of the AquaMMapS analysis for the “APO” trajectory (**left**) and for the prolonged SuMD trajectory of LUF5833 (**right**). The cells where the water molecules have an occupancy value higher than 25% are displayed as spheres coloured according to the occupancy value (from white to blue).

## Data Availability

Not applicable.

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
