# Peer review of "Ribose and Non-Ribose A2A Adenosine Receptor Agonists: Do They Share the Same Receptor Recognition Mechanism?"

_biomedicines, 2022, doi:10.3390/biomedicines10020515_

Round 1

Reviewer 1 Report

This simulation study seems to have been carried out competently and unravels the possibility that different classes of A2AR agonists may differently interact with A2AR. Furthermore, there is a possible molecular basis to understand the constitutive activity of the receptor, which the authors’ name as ‘basal activity’, which is mentioned en passant in a single sentence. Thus, the robustness of the methodology and the novelty of the data prompt a recommendation for the acceptance of this manuscript. Having said that, I would recommend the authors to consider inserting some words of caution on some aspects related both to methodology and to the insertion of this manuscript in its field, namely:

1-In terms of constraints imposed by the methodology, two important features of the functioning (and therapeutic utility) of A2AR should be considered: i) pH, since A2AR are engaged at acidic pHs and slight pH changes have major impact on A2AR function; ii) lipid constitution of the membrane where A2AR are imbedded, specially because cholesterol seems to profoundly affect A2AR function. I do not expect to see new data, but to insert a comment on these two important aspects (the first might be readily testable, but I wouldn’t know how to tackle the second).

2-A key aspect of A2AR signaling in the time of residency of the agonists. Apart from the path of interaction, some comments on this important aspect are warned.

3-It is somewhat mandatory to clearly highlight that the conception of an individual receptor in a defined lipidic and ionic environment is certainly of interest, but it cannot recapitulate important features of A2AR, namely their complex set of molecular interaction with itself and other GPCR (heteromers) and a pleora of ancillary factors (superiorly reviewed in Ijzerman et al., 2002, Pharmacol Rev, in press), which leads to surprising pharmacological properties in different brain areas as well as a different pattern of G-protein coupling, as best highlighted by Fredholm’s work (Cunha et al., 1996, Arch Pharmacol 353:261-71; Cunha et al., 1999, Arch Pharmacol 359:295-302). These limitations should be inserted as a final paragraph in the manuscript.

Author Response

Reviewer 1:

  • 1-In terms of constraints imposed by the methodology, two important features of the functioning (and therapeutic utility) of A2AR should be considered: i) pH, since A2AR are engaged at acidic pHs and slight pH changes have major impact on A2AR function; ii) lipid constitution of the membrane where A2AR are imbedded, specially because cholesterol seems to profoundly affect A2AR function. I do not expect to see new data, but to insert a comment on these two important aspects (the first might be readily testable, but I wouldn’t know how to tackle the second).

A paragraph regarding these two major limitations of the methodology has been added at the beginning of Materials and Methods (section 2.1: System Setup; page 3, lines 31-43).

  • 2-A key aspect of A2AR signaling in the time of residency of the agonists. Apart from the path of interaction, some comments on this important aspect are warned.

The issue has been addressed at the end of Results and Discussion (section 3; page 11, lines 16-21)

  • 3-It is somewhat mandatory to clearly highlight that the conception of an individual receptor in a defined lipidic and ionic environment is certainly of interest, but it cannot recapitulate important features of A2AR, namely their complex set of molecular interaction with itself and other GPCR (heteromers) and a pleora of ancillary factors (superiorly reviewed in Ijzerman et al., 2002, Pharmacol Rev, in press), which leads to surprising pharmacological properties in different brain areas as well as a different pattern of G-protein coupling, as best highlighted by Fredholm’s work (Cunha et al., 1996, Arch Pharmacol 353:261-71; Cunha et al., 1999, Arch Pharmacol 359:295-302). These limitations should be inserted as a final paragraph in the manuscript.

A paragraph regarding these limitations of the methodology has been added at the end of Results and Discussion (section 3; page 11, lines 3-16).

Reviewer 2 Report

This manuscript poses that simply understanding the bound state of a ligand to a GPCR does not necessarily explain its activation mechanism. Importantly, we need to understand the pathways of receptor recognition. The authors use structures of the A2A receptor and conduct (supervised) MD simulations to explore how four agonists recognize and bind the receptor. They find the agonist LUF5833 significantly differs in its binding mechanism.

Overall the manuscript is well presented. I suggest the following minor changes/edits:

Abstract: replace "renovated" with "renewed"

There are various places where a space is missing eg line 24 Gαiprotein. Manuscript needs checking.

I think throughout replace ARs with AR eg ARs agonists edit to AR agonist.

p3 line 2-3; stating protein-ligand binding is a rare event is ambiguous or unclear. I think you mean the mechanism of ligand-receptor recognition involves infrequent (rare) collisions and yet are essential for function. It needs to be edited/clarified.

on p4 line 36, the meaning of the "two centers of mass" is not clear; between what exactly? Perhaps to the MD expert it is clear; if this could be edited for clarification that would be helpful.

Overall the results and figures are clear and the idea of differences in recognition are clear. I thought discussion was under whelming. For instance you state in the introduction that mutagenesis has different effects for the various agonists. Could this be discussed in context to your findings especially after the sentence "This suggests a different
13 binding pathway for the two classes of ARs agonists." (p10 line 13). In other words are your findings consistent with the reported experimental work? Similarly, the proposal that bound water can account for basal activity (p10 lines 20-22) - does mutagenesis experiments support this proposal?

Author Response

Reviewer 2:

  • Abstract: replace "renovated" with "renewed"

The issue has been corrected in the manuscript

  • There are various places where a space is missing eg line 24 Gα Manuscript needs checking.

The issue has been corrected in the manuscript

  • I think throughout replace ARs with AR eg ARs agonists edit to AR agonist.

The issue has been corrected in the manuscript

  • p3 line 2-3; stating protein-ligand binding is a rare event is ambiguous or unclear. I think you mean the mechanism of ligand-receptor recognition involves infrequent (rare) collisions and yet are essential for function. It needs to be edited/clarified.

The issue has been corrected in the manuscript

  • on p4 line 36, the meaning of the "two centers of mass" is not clear; between what exactly? Perhaps to the MD expert it is clear; if this could be edited for clarification that would be helpful.

The issue has been corrected in the manuscript

  • Overall the results and figures are clear and the idea of differences in recognition are clear. I thought discussion was under whelming. For instance you state in the introduction that mutagenesis has different effects for the various agonists. Could this be discussed in context to your findings especially after the sentence "This suggests a different
    13 binding pathway for the two classes of ARs agonists." (p10 line 13). In other words are your findings consistent with the reported experimental work? Similarly, the proposal that bound water can account for basal activity (p10 lines 20-22) - does mutagenesis experiments support this proposal?

The Discussion section of the manuscript has been expanded, introducing a comparison of data coming from our SuMD simulations with available mutagenesis studies.